# Expert Recommendation for Answering Questions on Social Media

**Kyoungsoo Bok [1], Heesub Song [2], Dojin Choi [2], Jongtae Lim [2] , Deukbae Park [2] and Jaesoo Yoo [2],\***

[1] Department of SW Convergence Technology, Wonkwang University, Iksandae 460, Iksan 54538, Jeonbuk, Korea; ksbok@wku.ac.kr
[2] Department of Information and Communication Engineering, Chungbuk National University, Chungdae-ro 1, Seowon-Gu, Cheongju 28644, Chungbuk, Korea; heeusb@chungbuk.ac.kr (H.S.); mycdj91@chungbuk.ac.kr (D.C.); jtlim@chungbuk.ac.kr (J.L.); pdbdbp@naver.com (D.P.)
\* Correspondence: yjs@chungbuk.ac.kr; Tel.: +82-43-261-3230

**Featured Application: Because a variety of information is created and shared on social media, studies have been performed to identify experts who can provide necessary information to users. We recommend experts who can answer social users' questions on social media.**

**Abstract:** In this paper, we propose a method for recommending experts to appropriately answer questions based on social activity analysis on social media. By analyzing various social activities performed on social media, the user's interests are identified. Through the human relation analysis of the users of a particular interest field and by considering the response speed and answer quality of the user, we determine the influence of a user. An expert group is matched by analyzing the content of queries by a user and using a hierarchical structure of words. For a user question, the accuracy of an expert recommendation is enhanced by incorporating the question content and sublevel words based on the hierarchical structure of words. Various evaluations have demonstrated that the performance of the proposed method is superior to existing methods.

**Keywords:** social activity; expert; recommendation; user interests; human relation; answer quality

## 1. Introduction

The use of social media for generating and sharing thoughts or opinions among users has increased [1–4] with advancements in network technology and mobile devices. Social media are communication media that facilitate interactions or develop interdependent relationships between individuals and groups [5,6]. Social Networking Service (SNS) is a typical social media service used for communication and information sharing between users based on a human network of users [7–10]. Recently, SNS has been used to create, reprocess, and propagate new information, having evolved from a method of simple information sharing.

Because a large amount of information is reproduced and shared exponentially through SNS, users want to see only the information they demanded [11–13]. Furthermore, because unverified information is spread by non-experts, users must determine the level of trustworthiness of the information circulating on social media. In such an environment, social media users wish to find experts who can provide the desired information or answers to their queries [14–16]. Recently, services have been developed to answer questions by social media users [17–19]. There are online Question and Answer (Q&A) services such as Yahoo Answers, Naver Knowledge iN, and Baidu Knows, where users can ask questions and answer questions freely online. Recently, social Q&A services that combine social media and online Q&A services have been developed. Quora, for example, is an online social service that enables anyone to ask questions and comment on the answers that have been submitted by other users. On social media, users can get answers to questions through



Q&A services by registering feature problems and questions online. One problem with these Q&A services is that there may be inaccurate answers because many unspecified users can answer questions, and there may be no answers to specific questions. Further, since any unreliable user may answer a question, it may be difficult to obtain a high-quality answer. To solve these problems, it is necessary to identify and recommend experts who can answer the user's questions appropriately [20–22].

Studies have been performed on expert recommendation services [23,24] that recommend users who provide the requested information or provide a high-quality answer to a given question. Existing expert recommendation methods include recommendation methods based on direct input profiles of social media users and recommendation methods based on the analysis of user activities. The user profile method determines the similarity of user profiles with the question content and recommends a user who possesses the highest similarity. Because it simply compares a question and user profiles, the cost is low, but if the user profiles have not been updated or wrong information has been input with malicious intent, the precision of expert recommendations becomes low, which is a drawback [25–28]. To ameliorate this drawback of the profile-based expert recommendation method, studies have been conducted on methods of recommending an expert by analyzing user activity details [29–32]. This method determines the similarity of a user's question by considering activities such as the user's wr recommended. Such a method recommends an expert more precisely compared to the profile-based expert recommendation method. In [29], an expert recommendation method was proposed using dynamic profiles, which are generated by analyzing the social activities of users. This method considers the generated dynamic profile of a user with a question, and a user with the highest similarity is recommended as an expert. The method of recommending an expert by generating dynamic profiles demonstrates higher precision compared with the itings, comments, likes, and friend relationships. Subsequently, a user with the highest similarity is profile-based expert recommendation method. However, the dynamic profile-based method cannot recommend an expert in a different field if the user has no human relationships or has human relationships in only one field. In [32], a topic-specific contextual feature model (TSCFM) was proposed to recommend an expert for a topic by analyzing the social activities of users. This method recommends an expert for a topic by analyzing the social activities of users and demonstrates high precision when recommending an expert suitable for the topic. However, the quality of the answer cannot be guaranteed.

In this paper, we propose an expert recommendation method that considers user interest, human relationships, and answer quality. To solve the problem of the profile-based expert recommendation method, we extract user interest by analyzing their social activities. User's interests are determined by extracting fields that a user showed interest in recently, based on the social activities of the user. Since each user has different human relationships or relationships with different users, a specific expert cannot be recommended based on the user who asks the question. For the human relations of a user, the influence of a user is determined by analyzing how many relationships the user has formed with people of the same interest field using the human relations of users on social media. In the case of the expert recommendation method, not considering the answer quality of users, the response speed, or the answer quality of a user who has been recommended as an expert cannot be guaranteed. Answer quality is determined by analyzing how fast and how precisely a user answers another user's question. Furthermore, the method of matching an expert group is used by using the hierarchical structure of words for analyzing the question asked by a user. Based on the hierarchical structure of words for the user's question, the precision and trust of the expert's recommendation are improved by incorporating experts for the question content as well as sublevel words. To demonstrate the superiority of the proposed method, various performance evaluations were performed to compare with the existing methods.

This paper is organized as follows: Section 2 introduces relevant previous studies and methods for expert recommendation and presents their problems. Section 3 describes

in detail the characteristics and process of the proposed expert recommendation method. Section 4 proves the superiority of the proposed method through comparative performance evaluations between it and the existing expert recommendation methods. Finally, Section 5 provides the conclusions of this study and the directions for future studies.

## 2. Related Work

Ref. [17] proposes a method of searching experts who can answer a given question by extracting interest keywords through user profile analysis. Three search strategies are proposed: no-extension, cluster, and tree. The no-extension strategy compares a question and keywords on the user profiles. The cluster strategy constructs a common cluster and a question cluster and searches for profiles with keywords similar to those appearing in the question and of users who have a human relationship with the questioner, and subsequently expands the clusters by including synonyms. A similar keyword comparison is also made between the expanded common cluster and question cluster, and the user profile with the best match is recommended as an expert. The tree strategy constructs trees, using the Wordnet hierarchical structure, of keywords of users with human relations and those contained in a question. The tree structures of the question and keywords are compared, and a user showing a similar tree structure is recommended as an expert.

Ref. [29] proposes a method of searching an expert by creating user profiles based on the social activity analysis of users and using human relations on social media. A dynamic user profile is generated through the periodic analysis of recent activity by a user. Considering the reputation score and answer score of a user, the trust of a user is determined. A ranking is assigned according to their expertise level, and a user of high trust and precision is determined to be an expert.

Ref. [32] proposes a method of searching for an expert on a certain topic by analyzing users' social activities. The existing expert recommendation methods make recommendations based on interactions between users unrelated to the topic. The social activities of users on social media are analyzed, and the location and time data for users are collected from human relations and written documents. Subsequently, the topic recognition model obtains the characteristics of topics suitable for a question in the collected social activities of a user, and a context recognition model obtains the characteristics of a context suitable for the question from the user's social activities. Using the context characteristics model by topic, user scores are calculated, and an expert is recommended to the user.

The method of analyzing user profiles on social media yields a faster recommendation time compared with other methods. However, while the interest fields of users gradually change as time progresses, the majority of social media users do not update their profiles [17]. Because users do not update their profiles, the latest interest fields of the users cannot be extracted. Furthermore, some users write and input their profiles falsely. If an expert is recommended based on the profiles, a user who entered false information may be recommended as an expert. Consequently, the trust in the recommendation result is degraded. The method of recommending an expert by generating dynamic profiles of users exhibits higher precision than the profile-based expert recommendation method [29]. However, such an expert recommendation method cannot recommend an expert precisely when the number of users who have formed human relations with the user is insufficient or when the users are actively engaged in the field of interest. Furthermore, when the human relations of a user are formed with users of a similar field, users suitable for diverse queries of a user cannot be found. Ref. [32] proposes a method of recommending an expert for a topic by analyzing the social activities of users on social media. However, the expert recommendation method using the context characteristics model cannot guarantee the answer quality of an expert. A user can ask a question of an expert, and the expert can provide an appropriate answer for the question. However, if the expert recommendation service recommends a certain expert, and the expert does not provide an answer, the user will not be able to obtain an answer for the question.

## 3. The Proposed Expert Recommendation Method

### 3.1. Overall Architecture

The existing expert recommendation methods determine experts by analyzing the profiles or social activities of users. The profile-based expert recommendation method calculates the similarity between a question and profiles created by users and recommends a similar user as an expert. However, when profiles are not updated frequently or are written falsely, the precision of such expert recommendation methods decreases. The expert recommendation method based on the social activity analysis of users analyzes considers the expertise and up-to-datedness of interest fields. However, an accurate expert cannot be recommended to a user who exhibits a small range of human relations. Further, because the existing methods do not consider the answer quality, high-quality answers from experts cannot be guaranteed. To solve the problems of the existing expert recommendation methods, a method that recommends an expert for each field who can answer the question properly based on the analysis of social activities and human relations of users on social media is proposed herein. The proposed method considers the up-to-datedness of user interest fields by analyzing the social activities of users. Based on the analysis results for user interest fields, human relations are reconstructed, and interest fields are considered, thereby yielding a human relation. To consider the response speed and quality of an expert, the answer quality is determined by analyzing the user's answers.

Figure 1 shows the overall structure of the proposed expert recommendation method. The proposed expert recommendation consists of social activity collection, expert analysis, and expert recommendation modules. The social activity collection module collects various user activities such as posts, comments, the number of likes, shares, posting time, and human relations on social media to determine user expertise. The expertise analysis module calculates user interest, human relations, and answer quality through social activity analysis and determines the expertise score. To determine the user's interest, the keywords of the user are extracted from the user's activities. The user's interest is extracted using the posting time, the number of likes, the number of shares, and the number of comments for each extracted keyword. The human relationship is determined by analyzing each user's interest and human relations on social media and reconstructing the human relations for each interest field. Subsequently, the answer quality is determined using the response speed and answer trust of the user, as well as the similarity of the interest fields of the queries of other users, answering speed, and satisfaction of the questioners. The expert recommendation module recommends experts based on the expertise score calculated through the expertise analysis module. When a user asks a question, the keywords included in the question are extracted by analyzing the user's question. For the keywords extracted from the question, an expert is matched using the hierarchical structure of words, and the matched expert group is recommended.

### 3.2. User Interest

In social media, each user writes a profile when joining. This profile is used to recommend similar users, including the interesting fields or hometown of the user when starting the SNS. However, most SNS users do not update their initial profiles even when their interest fields change. When an expert is recommended using the initial profile of an SNS user, a problem occurs whereby a user with low expertise or outdated interests is recommended as an expert. Hence, in this paper, the changing interest field of users is determined by analyzing the social activities of users. If the social activities of a user are analyzed, their changing interest fields can be kept up to date. Furthermore, even when a user has composed a false profile, if their trust is low compared with the user activities, that user is not determined to be an expert. Thus, the precision of expert recommendation increases.

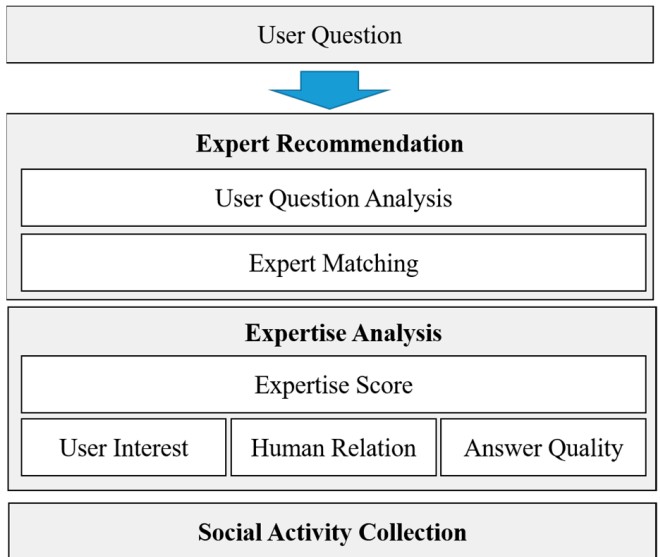

**Figure 1.** Overall architecture of the expert recommendation method.

Figure 2 shows the procedure of determining user interest. User interest is based on data shared or writing posted by a user on social media. The user's writings or shared data are interpreted as the expression of user interest. First, the user's trust is calculated for the social activities of the user. Social activity trust is determined by using the time of activity of the user, number of likes, number of comments, and number of shares. Using the morphological analyzer, nouns are extracted, and stopwords are removed. From the extracted words, important keywords or keywords frequently used by the user on SNS are extracted using the term frequency-inverse document frequency (TF-IDF) method. The user interest is calculated by adding the social activity trust and TF-IDF weight.

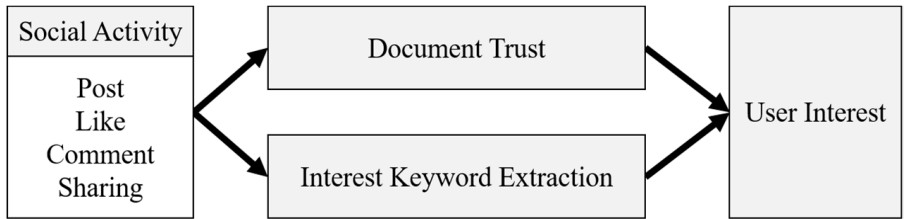

**Figure 2.** Determination procedure of user interest.

The TF-IDF weight used in the method of extracting keywords of interest field is a weight used in information search and text mining. When a document group composed of multiple documents exists, it is used as a statistical value exhibiting the importance of a certain word in a certain document. Particularly, it is used to extract keywords in a document. TF indicates the term frequency, i.e., how frequently a certain word appears in a document. IDF indicates the inverse document frequency and indicates the number of documents that contain a certain word in the document. The TF-IDF is obtained by multiplying the TF and IDF. For example, we assume that 100 documents exist that a user is engaged in. Among them, eight documents contain the word "Hadoop". We suppose that the word "Hadoop" appears five times in one of those documents. Subsequently, the TF value for that document is 5, and the IDF is 2.52573. Therefore, the TF-IDF value is 12.6286. To extract the user's interest, keywords are extracted using the TF-IDF method for the user activity details. The higher the TF-IDF value, the more important the keyword. It determines the trust of the document in which the extracted keywords appear.

User interests are expressed by the weight of keywords extracted from documents created through user social activities. The keywords with a high weight mean a high

interest of the user. Equation (1) is the weight of the keyword $i$ of user $k$, where $n$ is the number of documents left by user $k$, $TF_{ik}$ is the term frequency that represents the frequency appearing in the document, $DT_{jk}$ is the trust of document $j$ indicated by user $k$, and $IDF_{ik}$ is the inverse document frequency that represents the number of documents that contain a keyword in the total document.

$$KW_{ik} = \left( \frac{\sum_{j=1}^{n} TF_{ik} \cdot DT_{jk}}{n} \right) \cdot IDF_{ik} \tag{1}$$

The method of evaluating the trust of a document assigns a larger time weight to more recent writings by assigning a weight to the up-to-datedness of a document and determines the trust of a document by considering the document's number of likes, number of comments, and number of shares. Equation (2) calculates the document trust left by user $k$. $TW_{jk} = \sqrt{-0.002739t + 1}$ is the time weight of a document left by the user and has a lower weight when a long time has passed since the document was written. $LW_{jk}$ is the average value of likes of a document left by the user, $CW_{jk}$ is the average value of comments on the document left by the user, and $SW_{jk}$ is the average value of shares of the document left by the user.

$$DT_{jk} = TW_{jk} + LW_{jk} + CW_{jk} + SW_{jk} \tag{2}$$

To determine the user's interest, the keywords extracted from the documents are sorted in ascending order according to their weights. The user interest $UI_k$ of user $k$ is composed of Top-N keywords with high weight. However, if the user interest is calculated using keyword scores produced as such, a user of low score may be sometimes recommended as an expert in the case of the latest keyword compared with that of the oldest word because the quantity of documents is different between the keywords. To solve this problem, the user interest is calculated by dividing by the sum of user groups of the same interest on social media.

### 3.3. Human Relation

It is important to consider human relations in the expert recommendation method. An expert tends to form relationships with many people in the same field rather than people in diverse fields. The existing expert recommendation method employs a user matching the question by simply analyzing profiles or recent social activities. Expert recommendations that do not use human relations may recommend a user who has falsely and maliciously composed a profile or posted writings that have not been verified by other users. To solve this problem, the proposed method calculates the human relation by reconstructing and analyzing the human relations of users for each field of interest.

Figure 3 shows the procedure for determining human relations by the interest field. The method of determining the human relation analyzes the user activities on social media first and, using the results by extracting the interest fields for each user, the human relations are reconstructed by the interest field. Using the PageRank algorithm on the reconstructed human relations, the human relationship is determined for each field of interest. The PageRank algorithm was created to improve the quality of search engines and calculates the score per webpage through the link relationship of the webpage. When the PageRank algorithm is applied to human relations, an expert of one field obtains a high PageRank score because they have human relations with many users of the same field. Meanwhile, a user of low expertise has a low PageRank score because they have human relations with few users of the same field. In this paper, the PageRank algorithm is applied to human relations and is used for determining the human relation of a user.

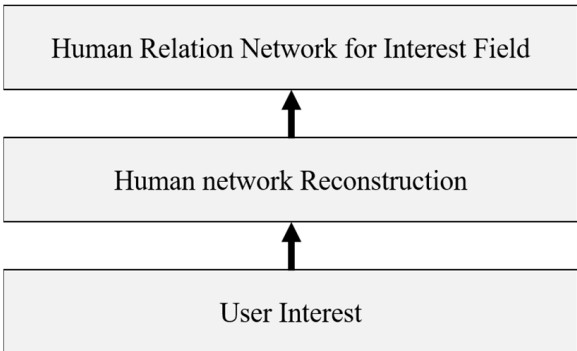

**Figure 3.** Procedure of determining human relation by field of interest.

Figure 4 shows the construction of the human relations for each field of interest. For example, we suppose the human relations of SNS are mixed in two other fields of interest called A and B. To calculate the human relations score for interest field A, interest field B is excluded, and the human relations are reconstructed using interest field A only. Subsequently, using the PageRank algorithm, the human relation is calculated. By the same method, the human relation is calculated for each field of interest.

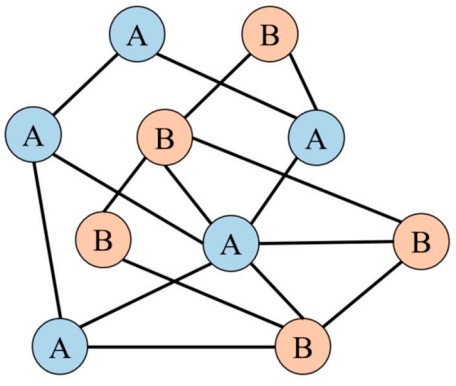

(**a**) Initial human relations.

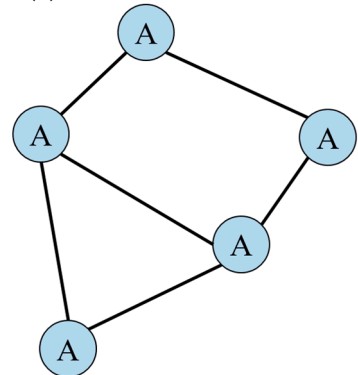

(**b**) Human relations of interest field A.

**Figure 4.** *Cont.*

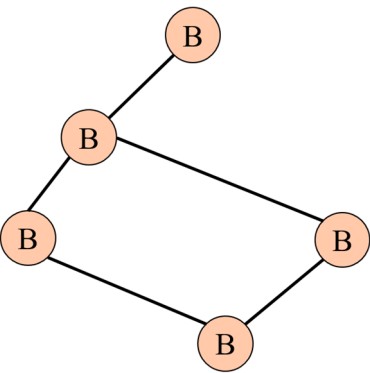

(**c**) Human relations of interest field B.

**Figure 4.** Human relation by field of interest.

The method of determining the human network for each interest uses the PageRank algorithm based on the human network reconstructed for each interest. Equation (3) is the human relation of user $k$ for particular interest $i$, where $n$ is the total number of users of interest $i$, $j$ is a friend of user $k$, and $d$ is a dumping vector of value between 0 and 1. $KW_{ij}$ is the weight of keyword $i$ for user $j$. $NU_{ij}$ is the number of friends of user $j$ in interest $i$.

$$R_{ik} = \frac{1-d}{n} + d \cdot \sum_{j=1}^{n} \frac{KW_{ij}}{NU_{ij}} \tag{3}$$

*3.4. Answer Quality*

An expert recommendation service allows a user to ask a question to a user recommended as an expert, and the expert can answer the question appropriately. However, if the quality of the answer is low, the user who asked the question cannot get a satisfactory answer. Even when the expert recommendation method recommends a knowledgeable expert suitable for the question of the user, if the expert does not answer, then user satisfaction is lowered. If the answer accuracy of the question is low, we can determine that the user who answered the question does not understand the question properly or is a non-expert. Therefore, we determine the quality of the answer considering the accuracy and satisfaction of the answer.

Figure 5 shows the procedure of determining the answer quality. First, the satisfaction of the answer is calculated by considering the answers by the experts, the number of likes to the answers, and the time required to answer. Next, the similarity between the answers and the interest field of the user is measured to determine the accuracy of the answer. Further, the answer quality is determined by combining the similarity of an expert's answer and the answer satisfaction.

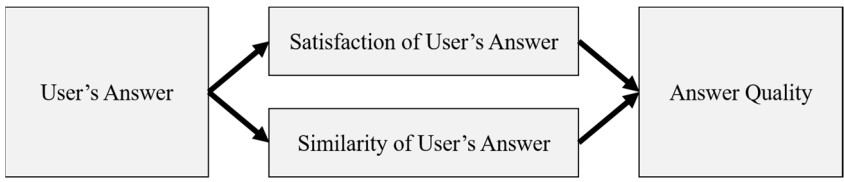

**Figure 5.** Procedure of determining the answer quality.

The answer satisfaction is determined by considering the amount answered by the user, response speed, the number of likes, and response time. Equation (4) is the answering quality of the keyword $i$ for user $k$, where $AAA_{ik}$ is the average number of answers to the question containing keyword $i$ for user $k$. $SA_{ik}$ is the answer similarity between the user interest and the user's answer, $ALA_{ik}$ is the average number of likes for user $k$, $ATA_{ik}$ is the average response time of user $k$. $AAA_{ik}$, $ALA_{ik}$, and $ATA_{ik}$ are calculated by the

Equations (5)–(7), where $Max(AA_i)$, $Max(LA_i)$, and $Max(TA_i)$ are the maximum number of answers, the maximum number of likes, and the minimum response time of keyword $i$.

$$AQ_{ik} = \frac{1}{2} \cdot (AAA_{ik} + SA_{ik} \cdot (ALA_{ik} + ATA_{ik})) \tag{4}$$

$$AAA_{ik} = \frac{AA_{ik}}{Max(AA_i)} \tag{5}$$

$$ALA_{ik} = \frac{1}{2} \cdot \frac{LA_{ik}}{Max(LA_i)} \tag{6}$$

$$ATA_{ik} = \frac{1}{2} \cdot \frac{TA_{ik}}{Min(TA_i)} \tag{7}$$

To determine whether the user's answer to a question is professional, the answer similarity $SA_{ik}$ is calculated. The answer similarity calculates the cosine similarity between the user interest and the keywords included in the answer. Cosine similarity measures the similarity level between two vectors of internal space using the cosine value of angle between two vectors. Because cosine similarity can be applied to any number of dimensions, it is often used for measuring the similarity in a positive space of multiple dimensions. Equation (8) is the cosine similarity for calculating the answer similarity, where $UI_k$ is the user interest vector of user $k$ and $AK_{jk}$ is a keyword vector included in answer $j$ written by user $k$.

$$SA_{ik} = \frac{1}{n} \sum_{j=1}^{n} \frac{UI_k \cdot AK_{jk}}{|UI_k||AK_{jk}|} \tag{8}$$

*3.5. Expert Recommendation*

An expertise score is derived by combining a user's interest, their human relations, and the answer quality. Equation (9) is the expertise score of each keyword $i$ for user $k$ by considering the answer quality and human relations, where $\alpha + \beta + \gamma = 1$. $UI_{ik}$ is the user interest, $HR_{ik}$ is the human relation, and $AQ_{ik}$ is the answer quality.

$$ES_{ik} = \alpha KW_{ik} + \beta HR_{ik} + \gamma AQ_{ik} \tag{9}$$

Using the expertise score calculated by Equation (6), an expert group is composed for each field that considers the interest fields, human relations, and answer quality of users. The method of composing an expert group is based on the ontology constructed based on the hierarchical relationship of words. Using the expertise score determined for each user, users are included as members of the ontology group.

The existing expert recommendation methods recommend an expert based on the results of matching with the search question requested by a user. However, when a question and an expert are matched, the precision of an expert recommendation decreases because the experts for the sublevel words of the question are not included in the recommendation result. The precision for the user's question is increased by including the experts included for the sublevel words by constructing hierarchical relationships of words, as well as the expert group for the question. Figure 6 shows the expert recommendation procedure. To match the user's question with the expert group, the keywords of the user's question are extracted by first analyzing the user's question. The extracted keywords from the user's question are matched with the hierarchical structure of the composed expert group. Subsequently, the matched expert group is selected. By determining the sublevel word, expert groups of matched expert groups, and including them in the matched expert group results, the final expert group is derived.

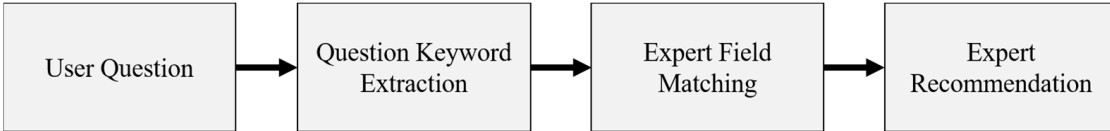

**Figure 6.** Procedure of expert recommendation.

The experts are recommended according to the expert's expertise score included in the expert group. If the word associated with the question has a lower level according to hierarchical relationships, the expertise score is reduced according to the sublevel number. For example, when a user question is matched with expert group B, as shown in Figure 7, expert group B and the sublevel expert groups C, D, and E of expert group B are selected as groups recommended as experts. Experts included in Group B use the values calculated in Equation (9) as expertise score. Experts included in the expert groups C, D, and E calculate the expertise score by reducing the expertise score by one-third.

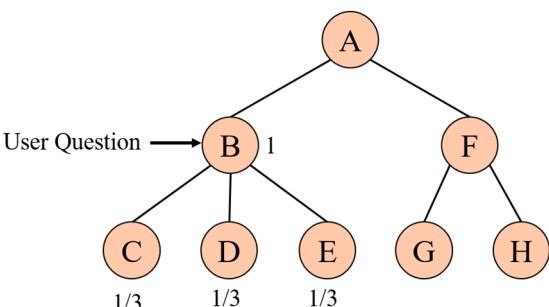

**Figure 7.** Example of matching a user question with expert groups.

## 4. Performance Evaluation

The superiority of the proposed method is verified through the performance comparison between the proposed expert recommendation method and the existing methods. First, a performance evaluation is conducted using the existing method developed by [29], which recommends an expert by analyzing the activity information of users. After generating dynamic profiles by collecting and analyzing the recent activities of each user, the method developed by [29] generates a question cluster using the ontology when a question is requested. Subsequently, by comparing the expanded question and the dynamic profiles of users, it recommends the most similar user as an expert. Herein, the method developed by [29] is defined as the Expert Finding considering Dynamic Profiles and Trust (EFDT). The second existing method is TSCFM [32], which analyzes the social activities of users on SNS and recommends a user suitable for the question and context by considering the user's human relations, location, and time. Since these two existing methods recommend an expert by analyzing the social activities of users, we choose them for the performance comparison.

The experimental evaluation was conducted in an environment of Intel Core i5 4440 @ 3.10 GHz, 8.00 GB Dual-Channel DDR3, and Windows 7 Ultimate K 64-bit. The analysis was performed for the expert recommendation results of using the proposed expert recommendation method. Moreover, an investigation was performed regarding how the recommendation result changed and the problems that occurred when the weights of user interest, human relation, and answer quality changed in the proposed method. For the experimental data, arbitrary user data were generated with normal distribution because the amount of significant data was small in the actual data. The experimental data are shown in Table 1. The number of users was 1000; the number of human relations of the user ranged from 0 to 200; the number of documents left by the user ranged from 0 to 100; the number of keywords extracted from a document was 3; the trust score of a document ranged from 0 to 1 point; the number keyword types was 100; the answer quality ranged

from 0 to 1 point. For each user, the numbers were generated randomly to satisfy the normal distribution. Additionally, the Hannanum morphological analyzer [33] developed by KAIST was used to extract keywords.

**Table 1.** Experimental data of performance evaluation.

| Parameter | Value |
|---|---|
| Number of users | 1000 |
| Number of human relations | 0–200 |
| Number of user documents | 0–100 |
| Number of keywords | 3 |
| Trust score of the document | 0–1 |
| Types of keywords | 100 |
| Answer quality score | 0–1 |

The proposed method cannot satisfy every user because the importance of the desired attribute is different for each user with respect to the interest fields, human relations, and answer quality of users. Moreover, it is difficult to set one overall weight as an important attribute. Therefore, in this paper, a different value is assigned to each weight to conduct the experimental evaluation and the recommendation results, owing to the weight value change changes. Table 2 shows the different values of weights $\alpha$, $\beta$, $\gamma$ used in Equation (9). The weight of $\alpha$ is the user interest, the weight of $\beta$ is the human relation, and the weight of $\gamma$ is the answer quality. When the number of the recommended experts is 20 among 100 persons, we measure the recall and precision by changing the weight values.

**Table 2.** Weights of user interest, human relation, and answer quality.

| Weight | C1 | C2 | C3 | C4 | C5 | C6 |
|---|---|---|---|---|---|---|
| $\alpha$ | 0.4 | 0.4 | 0.2 | 0.4 | 0.7 | 0.7 |
| B | 0.4 | 0.2 | 0.4 | 0.3 | 0.3 | 0 |
| $\gamma$ | 0.2 | 0.4 | 0.4 | 0.3 | 0 | 0.3 |

Figure 8 shows the recall and precision of expert recommendation results based on the weight values in Table 2. As a result of performing experimental evaluations using various weights, it was shown that the recall and precision were the highest when $\alpha$ = 0.4, $\beta$ = 0.4, and $\gamma$ = 0.2. It was shown that the probability of obtaining excellent performance in the expert recommendation result is high when a high weight is assigned to the user interest. Furthermore, the human relation demonstrated a higher probability of recommending an accurate expert compared with the answer quality of an expert.

When an expert was recommended by setting the expertise high in the user information, the precision was determined by verifying whether a certain generated user was recommended as an expert. In the experimental evaluations, we generated a user of the highest expertise in the dataset. We verified whether that user was recommended as an expert of the highest expertise. Subsequently, the precision was compared with those of the existing expert recommendation methods. Table 3 shows the expert recommendation results. To confirm whether the generated user was precisely recommended as an expert when the expert recommendation method was used, the expertise of a certain user was set to a high score. 100 users engaged in the SNS were used as experimental data. The information of generated user 10 was selected as an expert and set to the highest score. Because the proposed method recommended an expert by considering the interest fields, human relations, and answer quality of users, the generated user 10 was precisely shown for the ranking number 1. Meanwhile, for an existing method, i.e., the EFDT [29] method, user 10 was ranked number 7 because it did not include human relations. For the TSCFM [32] method, it ranked fourth because it did not expand the question, thereby demonstrating low accuracies.

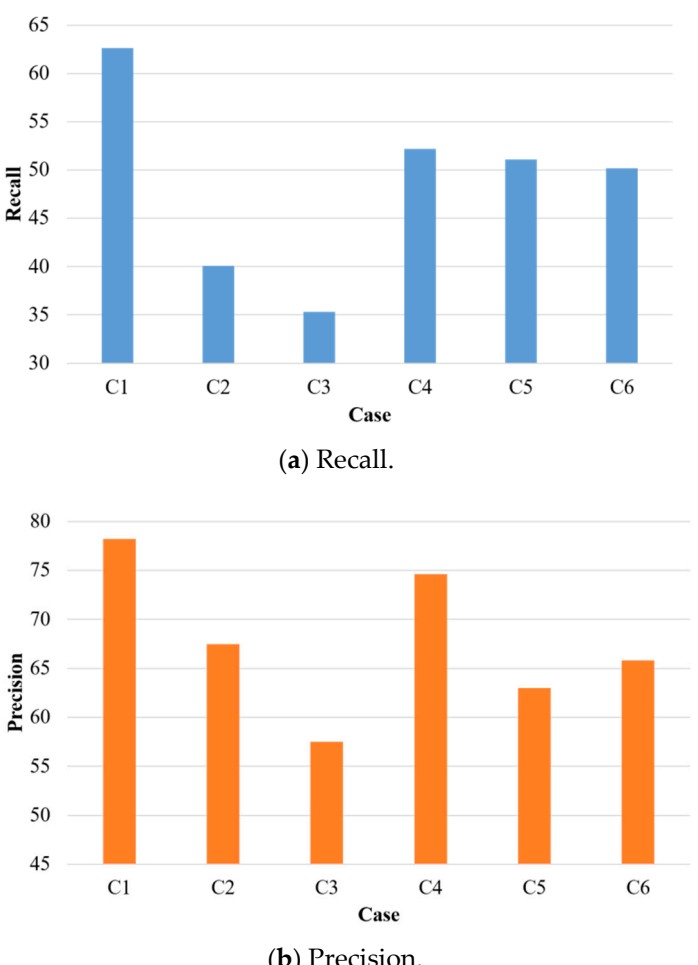

(**a**) Recall.

(**b**) Precision.

**Figure 8.** Recall and Precision according to weight value.

**Table 3.** Expert recommendation results.

| Ranking | Proposed Method | EFDT | TSCFM |
|---------|-----------------|---------|---------|
| 1 | User 10 | User 1 | User 2 |
| 2 | User 4 | User 7 | User 3 |
| 3 | User 7 | User 4 | User 13 |
| 4 | User 5 | User 14 | User 10 |
| 5 | User 9 | User 5 | User 8 |
| 6 | User 1 | User 9 | User 11 |
| 7 | User 13 | User 10 | User 9 |
| 8 | User 14 | User 19 | User 15 |
| 9 | User 19 | User 24 | User 18 |
| 10 | User 6 | User 27 | User 20 |
| 11 | User 23 | User 13 | User 22 |
| 12 | User 24 | User 6 | User 25 |
| 13 | User 18 | User 23 | User 28 |
| 14 | User 30 | User 30 | User 23 |
| 15 | User 27 | User 18 | User 1 |

We compared the proposed method with the existing methods in terms of recall, precision, and F-measure when the number of expert recommendations changes from 5 to 40. Recall is a proportion of users recommended as a real expert among the users classified as experts. The precision is the proportion of results classified as experts in the results recommended as experts. The F-measure is the harmonic mean of the precision and recall.

Figure 9 shows the recall according to changes in the number of expert recommendations. The proposed method showed higher recalls than the existing methods on average. All three methods indicated low precisions when five persons were recommended. In the proposed method, the recalls of more than 50% were shown when 10 persons or more were recommended. The proposed method and TSCFM exhibited higher recalls than the EFDT method. The reason was that human relations were determined in a larger scope. Furthermore, in the proposed method, since experts were recommended using the human relations, answer quality, and hierarchical structure of words in a question, higher recalls were indicated compared to the existing methods.

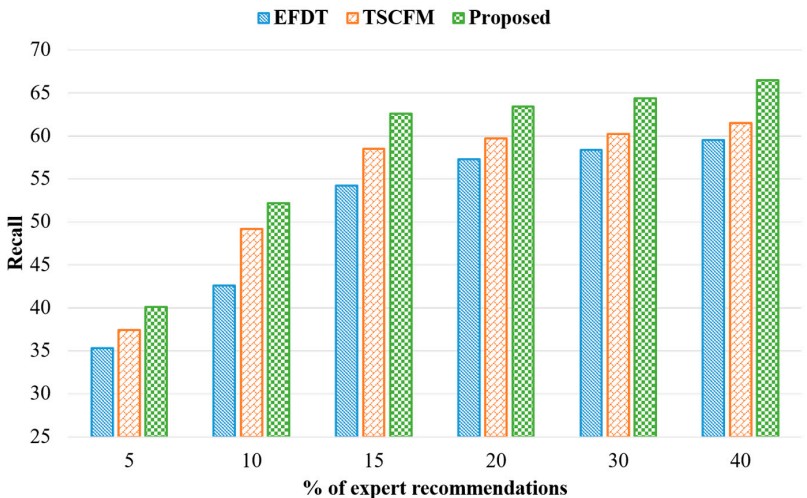

**Figure 9.** Recall according to the number of expert recommendations.

Figure 10 shows the precision according to the change in the number of expert recommendations. The proposed method showed about 8.1% higher precision than the existing methods, on average. The proposed method demonstrated a higher precision compared to the existing methods when five persons were recommended. Furthermore, better precision was continuously shown when more experts were recommended. The existing methods did not consider the human relations and answer quality of users. Furthermore, since the hierarchical relationship for the user question was not included, the expertise score was evaluated as low.

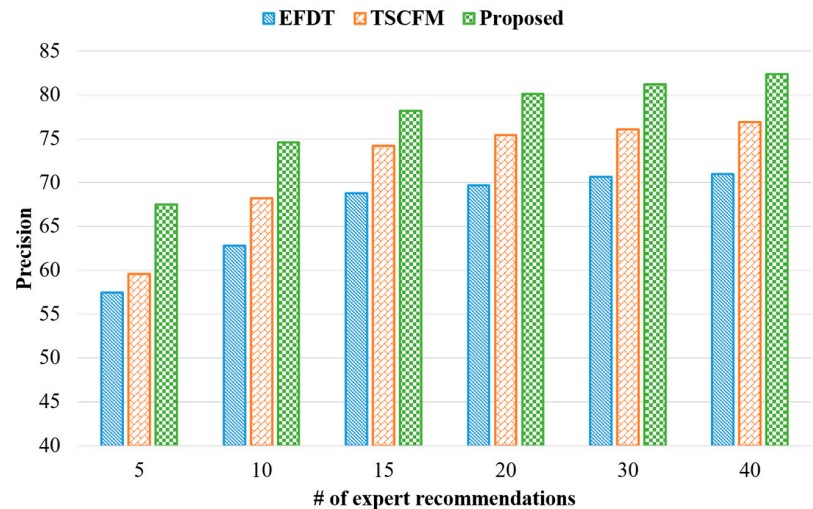

**Figure 10.** Precision according to the number of expert recommendations.

Figure 11 shows the F-measure according to the change in the number of expert recommendations. The proposed method showed about 6.5% higher F-measure than the existing methods, on average. The existing methods did not consider the human relations and answer quality of users. Furthermore, since the hierarchical relationship for the user question was not included, the real experts were not evaluated with high scores, and the expertise score was evaluated as low.

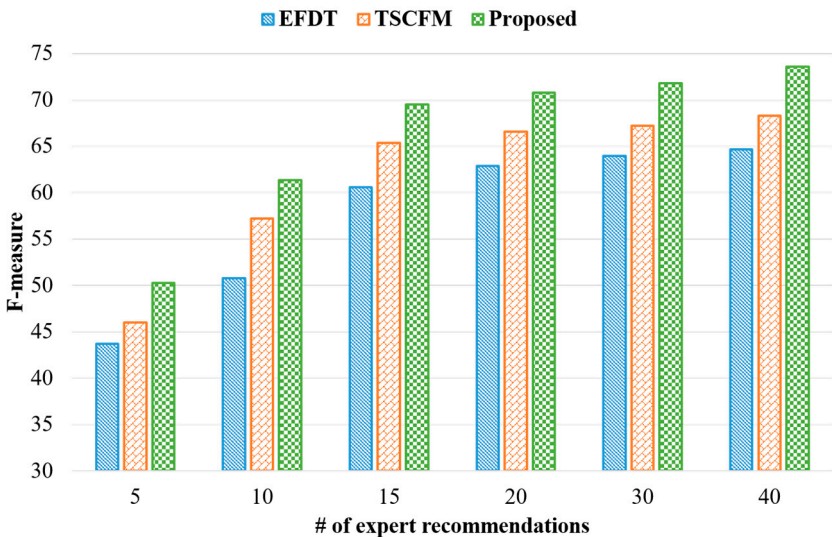

**Figure 11.** F-measure according to the number of expert recommendations.

## 5. Conclusions

We proposed a new expert recommendation method by considering the user interest, human relation, and answer quality on social media. The user interest was determined based on keywords and trust extracted by analyzing the user activities. The human relation was determined by the PageRank algorithm in the reconstructed interest network. The answer quality was determined by assessing how quickly and precisely the user had responded. The expertise score was determined by adding the user interest, human relation, and answer quality. To recommend an expert who can answer a user's question well, a keyword included in the user's question was extracted, and a matching expert was recommended using the hierarchical relationship of words. As a result of the experimental evaluation, it was proved that the proposed method had improved recall, precision, and F-measure compared with the existing methods. When the proposed method is applied to social Q&A services such as Quora, it is possible to construct an expert pool and improve the quality of answers through experts.

**Author Contributions:** Conceptualization, K.B., H.S., D.C., J.L., D.P., and J.Y.; methodology, K.B., H.S., D.P., and J.Y.; software, H.S.; validation, K.B., H.S. and D.P.; formal analysis, K.B., H.S., D.C., J.L., and J.Y.; data curation, K.B. and H.S.; writing—original draft preparation, K.B. and H.S.; writing—review and editing, K.B. and J.Y. All authors have read and agreed to the published version of the manuscript.

**Funding:** This work was supported by the National Research Foundation of Korea (NRF) grant funded by the Korean government(MSIT). (No. 2019R1A2C2084257, 2020R1F1A1075529), by the Institute of Information & Communications Technology Planning & Evaluation (IITP) grant funded by the Korean government(MSIT) (No.2014-3-00123, Development of High-Performance Visual BigData Discovery Platform for Large-Scale Real-time Data Analysis), and the MSIT(Ministry of Science and ICT), Korea, under the Grand Information Technology Research Center support program(IITP-2021-2020-0-01462) supervised by the IITP(Institute for Information & communications Technology Planning & Evaluation).

**Institutional Review Board Statement:** Not applicable.

**Informed Consent Statement:** Not applicable.

**Data Availability Statement:** Not applicable.

**Conflicts of Interest:** The authors declare no conflict of interest.

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
