# Peer review of "Expert Recommendation for Answering Questions on Social Media"

_applsci, doi:10.3390/app11167681_

Round 1

Reviewer 1 Report

A fascinating topic approached with diligent research!

The introduction is concise and clear and could be further expanded to included real examples from social media to further illustrated the relevance and significance of expert recommendation and recommenders. 

In the same way, the conclusion would benefit from an additional paragraph correlating the outcomes to recognizable examples in social media.

The results section is packed with crucial information and can be further broken down to present the data in a ways that is less conflated and more readable. 

The use of language needs some review to ensure this great research is presented in a way that validates the work to the full extent. For example at the beginning of the abstract, it is unclear if the "expert recommendation  method"  refers to a person as the sentence then continues with "who". If so, then the first sentence needs to be corrected accordingly. If not, then the use of "who" is incorrect. Spelling and grammar need review throughout the text. For example, the 'Architecture' section title is spelled incorrectly. 

Thank you and good luck with this interesting work!

Author Response

Dear Reviewer,

We would like to sincerely thank you for your attentive indications and good comments. Our paper is partially rewritten in order to revise and complement your comments. Please refer to the attached file about the detailed revisions.

Many thanks.

Best regards,

Jaesoo Yoo

Reviewer 2 Report

  1. Figure 1 may be needed to add interactions between the "Expert Recommendation" component and the "Expertise Analysis" component.
  2. It may need to add the other components too.
  3. The knowledge system may consider the qualify of knowledge. 

Author Response

(The authors gave the same response as above.)
